# Parametric Analysis of the Combustion Cycle of a Diesel Engine for Operation on Natural Gas

**Sergejus Lebedevas and Tomas Čepaitis ***

Marine Engineering Department, Faculty of Marine Technologies and Natural Sciences, Klaipeda University, 91225 Klaipėda, Lithuania; sergejus.lebedevas@ku.lt
* Correspondence: tomas.cepaitis@wsy.lt

**Abstract:** The publication research task is related to one of the solution aspects in reference to decarbonization of transport by transferring the operation of diesel engines to natural gas. The results of converted diesel engines into operation with dual-fuel (D-NG) without significant constructive modifications are focused on forecasting the energy efficiency parameters of in-service engine models and evaluation of the reserves improvement. This paper presents energy efficiency parameters and characteristics of the combustion cycle methodological optimization of high-speed 79.5/95.5 mm diesel engine with a conventional fuel injection system. Interrelations between the indicated efficiency ($\eta_i$), combustion cycle performance parameters (excess air ratio ($\alpha$), compression ratio ($\varepsilon$), degree of pressure increase in the cylinder ($\lambda$), maximum cycle pressure ($p_{max}$), air pressure ($p_k$), air temperature ($T_k$) after compression, etc.), and heat release characteristics were determined and researched. Directions of the optimization when the engines were operating in a wide range of load ($p_{mi}$) modes were also obtained: the low energy efficiency in the low-load mode were due to reduced heat release dynamics (combustion time increased up to 200° CA). The main influencing factors for $\eta_i$ were the pilot-injection portion phase ($\varphi_{inj}$) and $\alpha$, optimization of $\varepsilon$ was inefficient. To avoid exceeding the permissible limits of reliability for $p_{max}$, the realized reserve of $\eta_i$ increase was estimated as 10%. Methodological tools for the practical application of parametric analysis to the conversion of diesel to dual-fuel operation have been developed and adapted in the form of a numerical modeling algorithm, which was presented in nomogram form. For improvement of initial energy parameters for a specific engine models heat release characteristics identification, accurate methods must be used. The proposed methodology is seen as a theoretical tool for a dual-fuel conversion models for in-service engines and has benefit of a practical use of a fast application in the industrial field.

**Keywords:** diesel engine; dual fuel; combustion cycle; energy efficiency; heat release characteristics; phenomenological MM; parametric analysis

## 1. Introduction

### 1.1. Background

The energy consumption of the transport sector represents approximately 31% of the total global energy balance. This sector is also a major polluter, being responsible for 25% of greenhouse gas (GHG) emissions and 40% of the nitrogen oxides ($NO_x$) and solid particulate matter (*PM*). Under the European Union (EU) 2020 strategy, Directive 2009/28/EC of the European Parliament and of the Council established the following targets: a 20% reduction in the energy consumption and the use of renewable sources to provide 20% of the energy consumed. The most recent EU strategy, MEMO/11/197, projects a 60% reduction in the consumption of fossil fuels originating from petroleum and a 40% reduction in the use of GHG-intensive fuels in aviation as well as a 40% reduction in maritime emissions. Biofuels and liquefied natural gas (LNG) are considered the most promising alternative fuels for maritime transport, with LNG being the best option to meet

the International Maritime Organization (IMO) Tier III ($NO_x$ emission control standard for vessels) requirements over a 20-year perspective [1]. Reduction of $CO_2$ greenhouse gas emissions by 20–25% is equally important, which contributes to the achievement of the promising IMO standard (Resolution MEPC.203(62)), in terms of the Energy Efficiency Design Index (EEDI). Natural gas (NG) is being applied in several sectors, particularly heavy transport, such as shipping and rail [2,3]. Projections predict a 25% growth in road transport using NGs, and an increase of 33% in passenger flows and 25% in freight in Europe in 2030 [4].

Transport diesel fleets with a wide range of different models could be converted to dual-fuel operation. Potential solutions for engine conversion for diesel-natural gas (D-NG) operation have been investigated and successfully implemented in "modern" engine models. Many heavy-duty vehicles are available in the market, and marine engine manufacturers offer a considerable number of new compression ignition engines that have a modern accumulator-type fuel system running on pilot batches of diesel and NG, such as Wartsila, MAN Diesel & Turbo, and IVECO. Diesel to gaseous fuel conversion of the popular diesel engines (DEs) leads to a reduction in $NO_x$ and $CO_2$ emissions by 90–85% and 20–10%, respectively, and almost completely eliminates *PM* and sulphur oxides from the exhausts [5–8]. Approximately 30% of marine power plants in the world [9,10], and 30% of "small energy" models [11,12], are equipped with a conventional fuel supply system, which is characterized by optimization constraints of high reaction fuel (HRF) injection phases [13–15].

According to AVL, the global leader in DE design and research, the lack of methodological principles for conversion and engine management strategies for different types of engine models and relevant for companies specialising in conversion of DE to dual-fuel operation is a crucial gap in the efforts to improve the ecological and energy efficiency potential of power plants [6,16–18]. In particular, this applies to the Eastern European region and Lithuania [9,19]. The efficiency of the engine cylinder combustion cycle is determined by the HRF time of the pilot portion. Its volume distribution in the cylinder is achieved exclusively through the accumulative supply system and the supply of gaseous fuel at a high pressure of 150–300 bar [8,20]. The environmentally effective technological features of modern dual-fuel engines, primarily high-pressure (up to 2000–3000 bar) fuel injection systems, mainly common rail (CR), are infeasible in still-operating DE fleet models with conventional fuel injection systems converted to dual-fuel operation [8,20].

One of the most common methods to improve energy and environmental performance is to advance the injection phase of the pilot HRF fuel portion. However, a significant limitation is the increase in the maximum cycle pressure ($p_{max}$) of the combustion cycle, reducing engine reliability. However, in this context, many research results indicate the favorable conditions for the application of this technology to a dual-fuel engine.

### 1.2. Subsection Influence of Dual-Fuel Operation on the Maximum Cycle Pressure

Papagiannakis et al. [21] investigated the influence of the mass fraction of NG on the combustion characteristics at three engine loads and speed modes. The test indicated that changing the mass fraction of NG from 0% to 80% resulted in considerably longer ignition and combustion times and lower cylinder pressure when the engine was running in the dual-fuel mode. By increasing the ratio of gaseous fuel, the cylindrical pressure in the dual fuel decreased, and the ignition and combustion times of the fuel increased. Abdelaal et al. [22] compared the combustion characteristics of a dual-fuel D-NG engine at a fixed speed of 1600 rpm, maintaining a fixed 20% diesel pilot portion size and varying the amount of NG supplied to the cylinder. Comparing the dual-fuel and DE modes, the cylinder pressure decreased by 6.7 and 6.2% after the combustion advanced by crankshaft rotation angles (CAs) of 2.8 and 5.5° CA, respectively. The rate of maximum increased pressure decreased by 0.39 and 1.14 bar/CA at 53 and 87% loads, respectively. The maximum cycle temperature was also lower. Exhaust gas recirculation (EGR) was also tested in the study; during the test, the $p_{max}$ decreased, and fuel ignition time increased, and these

parameters only deteriorated with increasing EGR. A decrease in $p_{max}$ of approximately 10 bar and an increase in the rate of pressure of 2.3 bar/° CA were observed in a previous study [23]. Studies on the influence of NG combustion characteristics in a dual-fuel engine [21,24] found that the pressure in the cylinder and the characteristic combustion phases of the working mixture were longer than those of a DE.

However, conflicting results were obtained for the $p_{max}$ change [21] in one study, in which $p_{max}$ was always low regardless of the load. Lounici et al. [24] found that the $p_{max}$ value was lower only when the engine was running at low load, and the $p_{max}$ of a high-load working dual-fuel engine was higher than that of a DE. This was a result of the improved gaseous fuel combustion and faster heat release.

In order better understand the combustion process and the mechanisms of natural gas/diesel dual fuel formation in a single cylinder-caterpillar 3400 heavy duty 137.2/165.1 mm engine, engine experimental studies were compared with numerical simulations performed using AVL FIRE v2014 software in conjunction with the sub-model CHEMKIN. Studies with a single-cylinder accumulative CR system 137.2/105.1 engine identified two different physical mechanisms of the combustion working mixture, depending on the high cetane number of HRF [25]. The tests were performed at a 25% partial load of the engine, with the relative part of the gas phase and an energy value of δ NG = ~75% (δ NG is the co-combustion ratio of NG (CCR NG)). When a delayed diesel injection time (DIT) does not exceed 30° CA before the top dead center (CA BTDC), the ignition and combustion dynamics characteristic of the DE operation are observed; free OH radicals cluster close to the HRF torch, thereby initiating intense combustion through a kinetic mechanism, with a further transition to flame diffusion transfer to the peripheral zones of the combustion chamber (CC). As a result, a relatively small increase in the ignition delay period ahead of the DIT 4–5° CA BTDC provides $p_{max}$ and maximum combustion temperature ($T_{max}$), leading to an increase in $NO_x$ emissions and in the effective efficiency ($\eta_e$). In the investigated DIT range of 30–50° CA BTDC, the thermodynamic conditions of the working mixture in the cylinder become insufficient for rapid ignition of the HRF. The doubling of the ignition delay period outweighs the increase in the high cetane number, returning the combustion process to the top dead center (TDC). Active ignition OH centers include a significant volume of the CC, and the combustion of the working gas–air mixture becomes single-phase. The local temperature field of the CC equalizes, thereby reducing $NO_x$ emissions. The parallel reduction in CC in the non-combustion peripheral zones is also favourable for the reduction in *CO* and *PM* emissions from incomplete combustion products. Studies that used high-speed 85/90 mm CR engines, with a DIT range from 50 to 5° CA BTDC, obtained qualitatively analogous results [26]. The peculiarities of the influence of a higher DIT on the ecological and energy characteristics of the engine have also been confirmed.

We supplemented the research by varying the HRF injection pressure parameter in the range of 500–1000 bar, while optimizing the DIT angle, dividing the HRF injection into separate stages, and optimizing the injection pressure value. The injection of HRF in earlier phases is effective for improving the energy ecology and efficiency of dual-fuel engines. A single-cylinder marine medium-speed diesel Wartsila 20 DF 200/280 mm with a CR system experiment was also investigated [20], to determine the influence of the distribution and structure of the HRF and the thermodynamic conditions of the compression process in the cylinder on the ignition of the HRF and the combustion process of the working mixture. The tests were performed at the average nominal operation of the engine load (BEMP = 19 bar) at a constant speed of $n$ = 1000 rpm. The HRF injection pressure varied from 1300 to 2100 bar. The tests of the engine were supplemented with diesel injection simulation models in order better understand local combustion conditions. The results show the importance of fuel distribution in the cylinder as a way to control the combustion stages and therefore its effect on combustion instability, exhaust emissions, and cycle dynamics. The DIT ranged from 15 to 50° CA BTDC. The high HRF ignition reactivity at the DIT range of up to 30° CA BTDC, measured by the equivalence ratio, $\beta$, of the locally used air utilization factor (otherwise, according to the HRF "equivalence ratio"), correlates with the results

of the research. The predominant range of $\beta$ (0.2–0.8) results in a relatively short ignition delay period of 10–17° CA and intense combustion. The subsequent DIT is accompanied by a noticeable decrease in the $\beta$ = 0.2–0.8 fraction owing to the combustion process of the homogeneous working mixture at the TDC. The combustion pattern 'softens', local temperature fields are balanced, and $NO_x$ emissions are reduced. In addition, changes in the thermodynamic parameters of the working mixture in the cylinder (delaying the closing phase of the intake valves) are also effective in increasing the ignition delay period of the HRF. The range of the required DIT determination in this case is significantly narrowed. In studies performed from 45° to 32° CA BTDC, $NO_x$ emissions were reduced to 0.5 g/m$^3$, which is the value regulated by MARPOL 73/78 Tier III. A similar result was obtained in studies with a six-cylinder 112/132 mm engine [27] with a CR system, advancing the DIT to 32° CA BTDC (δ NG of 90%) in load mode. In contrast, emissions of hydrocarbons ($CH$) increase, and $\eta_e$ decreases [21,26–29].

In terms of the scientific aspect and practical application, to improve the effect of energy and ecological parameters in the conversion of DE to D-NG fuel, it is necessary to identify methods for implementing the conversion. Thus, the potential related to the parameters, characteristics of the combustion cycle, and the composition of dual-fuel D-NG should be studied. However, statistics from research on the development of NG technology show that the lack of open access methodological solutions is a major obstacle [18,30]. Thus, by revealing methods for improving the energy performance of dual-fuel engines and applying mathematical modelling (MM), we can develop methodological principles for forecasting and improving the energy performance of DEs to dual-fuel tools for practical use. In parallel with the experimental methods [31,32], the parametric analysis of DEs can also be performed using MM methods [20,33–36]. Multidimensional MM is preferred among the numerical models available and is effective at solving systems of moment (Navje-Stokes), energy (Furje-Kirghof), and mass-endurance equations (e.g., FIRE (AVL, Austria), KIVA (Laboratory of Energy Research, Los Alamos, NM, USA), and VECTIS (Ricardo, Shoreham-by-Sea West Sussex, England)) when implemented in software packages. One of the most common MM benchmarking sources is the computational fluid dynamics model [37–39]. Despite the advantages of multidimensional MM methods, their use at the preliminary estimation and prognosis stage is hindered by the absence of technical data on the study object; these data are necessary for reliable modelling. Thus, a rational solution is the method of parametric analysis, which is widely adapted in combustion engine practice, focused on the initial stages of burning or modernization of new technologies [40–42]. Without specifying the design parameters and features of the research object, these methods operate and establish interrelationships between the energy efficiency, in particular, the efficiency, and the main parameters of the combustion cycle. Thus, the study of methods is based on a specific model but is flexible in assessing alternatives to its development, i.e., the main principle of the initial sketch design phase is achieved. In this context, the research results of parametric investigation on combustion and emissions characteristics of a dual fuel (natural gas port injection and diesel pilot injection) engine using 0-D SRM and 3D CFD approach, are significant, rationally combining the use of different research models. In this study, the combustion and emission characteristics of a dual-fuel engine were examined numerically. The zero-dimensional stochastic reactor model (0-D SRM) method was used for numerical analysis. SRM was validated experimentally and parametric analysis of a dual-fuel 6-cylinder 90/112 mm engine was used, varying performance such parameters as fuel mixture ratio, engine speed, and exhaust gas recirculation rate. The numerically obtained 0-D SRM results were in good agreement with the experimental results. The combustion process in the dual fuel engine was also analyzed using computational fluid dynamics (CFD) model using a commercial 3-D CFD engine simulation tool 'STAR-CD' in conjunction with mesh generator 'es-ice'. The evolution of the spray and the combustion process were analyzed in the engine cylinder sector in order to reduce the computation time during the simulations. This approach has been adopted as the basis for the research presented in the authors' paper. In this context, there is also a lack of data on engines with

a conventional fuel injection system converted to dual-fuel D-NG. Most of the research has been performed with engines with a multi-stage accumulator-type CR system, which is characterized by a wide range of possibilities to regulate and improve the characteristics of the combustion cycle.

Klaipeda University and Vilnius Gediminas Technical University (VGTU), with a conventional fuel system, perform complex analytical, experimental, and numerical MM to investigate the energy and ecological indicators of convertible dual-fuel DE using conventional fuel injection systems [43–45]. Among other things, comparative experimental studies of the energy and environmental performance of engines with diesel and dual-composition D-NG fuels were presented [43]. The results of the experimental data on heat and exergy balance are presented and analyzed in the publication [45]. The research presented in this publication is based on the complex experimental research performed by authors, which have been adapted towards methodological solutions. This paper presents the results of a study that aimed to develop methodological principles for appropriate forecasting and improving the energy performance of an in-service DEs engine with a conventional fuel supply system to dual-fuel D-NG engines and tools for their practical use. In the initial conversion phase of an existing model, it would be rational to introduce its expected energy potential to run on dual D-NG fuel without significant constructive changes. For this purpose, it would be rational to develop and adapt methodological tools for the operational use, especially for Industrial, which would become the goal of the performed research. It has been hypothesized that the methods for optimizing the energy efficiency parameters of a diesel engine duty cycle will also be effective in converting models running on dual D-NG fuel with a convection fuel injection system. For this purpose, we aimed to:

- Identify and investigate the parameters of the combustion cycle, the characteristics that determine the in-service energy performance of a dual-fuel D-NG engine, and their interrelationships;
- Develop, adapt, and provide development directions, practically methodological tools for the rational selection and optimization of combustion cycle parameters to achieve energy efficiency of the dual-fuel D-NG engine.

## 2. Methodological Aspects of the Research

The results of experimental and numerical studies performed by the authors in a previous work, described in detail in [43,45], were used as the basis for the parametric analysis of the combustion cycle.

### 2.1. Experimental Research

Direct injection Audi 1.9 turbocharged direct injection (TDI) engine tests were performed at the Internal Combustion Engines Laboratory of the Automobile Transport Department, Faculty of Transport Engineering, VGTU. A 1.9 TDI DE (1Z type), with an electronically controlled BOSCH VP37 distribution-type fuel pump and turbocharger used for the tests. The EGR system was disabled during the tests. The main parameters of the engine are listed in Table 1.

The supply of NG for the dual-engine operation was performed at low pressure to the engine air intake manifold using commercial gas equipment. This method of NG supply was chosen because of the minimal intervention required for engine reconstruction, without changing the possibility of running the engine only on diesel and maintaining the ecological and energy indicators specified by the manufacturer, and owing to the prevalence of this equipment in the market. A schematic diagram of the experimental test engines is presented in Figure 1. A KI-5543 engine brake stand was used for the load $M_B$ and crankshaft speed determination. The torque measurement error was $\pm1.23$ Nm. The hourly fuel consumption $B_f$ was measured by SK-5000 electronic scales and a stopwatch, and the accuracy of the $B_f$ determination was 0.5%. The NG fuel was measured by a Coriolis-type mass flow meter. The fuel flow meter was RHEONIK RHM 015 (see Figure 1

pos. 25), connected into the high-pressure fuel supply system before the gas reducer, which reduced the gas to a pressure of 1.5 bar. The flow meter measuring range was 0.004 to 0.6 kg/min with a high measurement accuracy of ±0.10%. The ranges of the measured parameters of engine operation and their accuracy, in accordance with the requirements of the applicable standards, are detailed [43].

**Table 1.** Engine specification.

| Parameter | VW-Audi 1Z 1.9 TDI |
|---|---|
| Displacement (cm$^3$) | 1896 |
| Bore × stroke (mm) | 79.5 × 95.5 |
| Maximum power (kW/rpm) | 66/4000 |
| Maximum torque (Nm/rpm) | 180/2000–2500 |
| Cooling type | Water cooling |
| Fuel supply system | Direct injection |
| Cylinders | 4 in line |
| Compression ratio | 19.5:1 |
| Aspiration | Turbocharge |

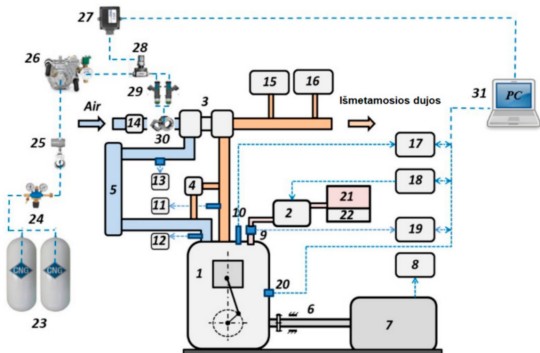

**Figure 1.** Schematic of the engine testing equipment.

### 2.2. Fuel Specification

Two fuel types were used during the experiment: liquid and gas. During the dual-fuel mode, standard diesel fuel (EN 590) and standard compressed NG (ISO 6976:1995) were used (see Table 2).

**Table 2.** Fuel properties.

| Fuel Property | Natural Gas | Diesel |
|---|---|---|
| Density (kg/m$^3$) | 0.74 | 829.0 |
| Cetane number | - | 49 |
| Lower heating value (MJ/kg) | 51.7 | 42.8 |
| Viscosity (cSt 40 °C) | - | 1.485 |
| H/C ratio | - | 1.907 |
| Component (% vol.) | Methane: 91.97 | Carbon: 86.0 |
|  | Ethane: 5.75 | Hydrogen: 13.6 |
|  | Propane: 1.30 | Oxygen: 0.4 |
|  | Butane: 0.281 |  |
|  | Nitrogen: 0.562 |  |
|  | Carbon dioxide: 0.0 |  |

1—1.9 TDI engine; 2—high-pressure fuel pump; 3—turbocharger; 4—EGR valve; 5—air cooler; 6—connecting shaft; 7—engine load plate; 8—engine torque and rotational speed recording equipment; 9—fuel injection timing sensor; 10—cylinder pressure sensor; 11—exhaust gas temperature meter; 12—intake gas temperature meter; 13—air pressure

meter; 14—air mass meter; 15—exhaust gas analyzer; 16—opacity analyzer; 17—cylinder pressure recording equipment; 18—fuel injection timing control equipment; 19—fuel injection timing recording equipment; 20—crankshaft position sensor; 21—fuel tank; 22—fuel consumption measuring equipment; 23—compressed natural gas tank; 24—pressure regulation valve; 25—gas flow meter; 26—pressure reducer; 27—ECU; 28—gas metering valve; 29—gas injectors; 30—air and gas mixer; 31—computer.

The experimental engine efficiency and emission research was conducted with a wide range of loads—(BEMP) brake effective mean pressure ($p_{me}$) and with engine speed $n = 2000$ rpm, as well as various HRF injection timing angles ($\varphi_{inj.}$). In every mode, characterized by different combinations ($p_{me}$, $\varphi_{inj}$), the engine parameters were measured using diesel only (D), and dual D and NG fuel: D60-NG40, D40-NG60, and D20-NG80 (here, the numbers following "D" and "NG" correspond to the diesel and NG percentage parts of the total energy balance). The engine load modes were named in the following manner: BEMP = *6 bar* ($p_{mi} = 8.2$ *bar*), high load mode (HLM); BEMP = *4 bar* ($p_{mi} = 6.2$ *bar*), medium load mode; and BEMP = *2 bar* ($p_{mi} = 4.2$ *bar*), low load mode (LLM).

### 2.3. Mathematical Model of Numerical Research on the Engine Combustion Cycle

The single-phase mathematical model was used in the research of engine energy parameter simulations and was implemented with software "IMPULS". The IMPULS program, developed at the Central Diesel Research Institute in St. Petersburg [46], has been successfully used in the development and modification of high-speed transport engines (150/150 mm, 150/180 mm, 165/180 mm, other) [47–49]. In the basic version, there are 18 sub-models. The software simulates a closed energy cycle DE model with a turbocharger. This is based on quasi-static equations of thermodynamics and gas dynamics, considering the exhaust system design parameters, variable gas turbine and compressor efficiency coefficients, heat losses to the engine cooling system, and environmental air parameters. The processes in the engine cylinder are described by a system of differential equations consisting of the laws of energy (1) and the mass (2) state Equation (3):

$$\frac{dU}{d\tau} = \frac{dQ_{re}}{d\tau} - \frac{dQ_e}{d\tau} - p \cdot \frac{dV}{d\tau} + h_s \cdot \frac{dm_s}{d\tau} - h_{ex} \cdot \frac{dm_{ex}}{d\tau}, \ [kJ/s], \tag{1}$$

$$\frac{dm}{d\tau} = \frac{dm_s}{d\tau} + \frac{dm_{inj}}{d\tau} - \frac{dm_{ex}}{d\tau}, \ \left[\frac{kg}{s}\right], \tag{2}$$

$$\frac{dp}{d\tau} = \frac{m \cdot R}{V} \cdot \frac{dT}{d\tau} + \frac{m \cdot T}{V} \cdot \frac{dR}{d\tau} + \frac{R \cdot T}{V} \cdot \frac{dm}{d\tau} - \frac{p}{V} \cdot \frac{dV}{d\tau}, \ [Pa/s], \tag{3}$$

where $U$ is the internal energy (J); $Q_{re}$ and $Q_{ex}$ are the heat release and heat exchange energies (J), respectively; $p$ is the pressure (Pa); $V$ is the volume (m³); $h_s$ supply working body enthalpy (J/kg); $h_{ex}$ exhaust working body enthalpy (J/kg); $m$ is the total mass (kg); $m_s$ is the supply air mass (kg); $m_{inj}$ is the mass of fuel sprayed (kg); $m_{ex}$ is the mass of exhaust gas (kg); $\tau$ is the time (s); $R$ is the gas constant (J/kg·K); and $T$ is the temperature (K).

The structure of this software is constantly improving and supplemented by sub-models of the working fuel mixture in the cylinder formation and combustion, assessing the dynamics of fuel injection, evaporation, flame spread; use of fuels with different chemical elemental composition, etc. Most of the phenomenological sub-models implemented in the program are similar to the other widely used software AVL BOOST [50]. The heat release was determined by the I. Vibe model [51] with G.Woschni additions [52,53], which are widely used in combustion engine combustion cycle modeling studies [54]. G. Woschni's analytical solutions ensure that I. Vibe heat release model parameters $m$—form factor and $\varphi_z$—conditional duration calibration mode are recalculated for different part load modes (changes in environmental conditions, design parameters) from indicators influencing the realization of the combustion cycle (excess air coefficient, fuel combustion induction period, pressure and temperature, working mixture, etc.) [53]. This includes declaring the compatibility of these solutions for different fuels, including gaseous fuels. The choice is

determined by the models of the targeted operating diesel engine fleet with a conventional fuel injection system. For the most part, the lack of initial data for the mathematical modeling of the parameters of these models (primarily related to the fast-moving physical processes of cylinders) leads to the application of simplified phenomenological O-D models in the initial phase of the study. Calibration of the parameter $m$ and $\varphi_z$ for the vibe model is usually performed based on experimental indicator diagram analysis results. However, the Bulati-Glanzman [55] method is also quite effective, and its application to the in-service engine does not cause significant difficulties in operating with the main energy parameters of the engine. The heat exchange in the engine cylinder is described by G. Woschni's equations separately for the piston, cylinder head, bushing surface in terms of the intensity of the macro-motion of the created air. The constants of the heat exchange calculation equations were calibrated according to the experimental data of the heat balance of the engine. In part load modes, the characteristic parameters of the turbo-charge system units (compressor, turbine, chiller) changes are modeled according to the respective analytical dependencies or "maps" data. The TEPLM software, used for the experimental indicator diagram analysis [56], uses a closed thermodynamic cycle energy balance model to evaluate the heat transfer through the cylinder walls. Initial values for the software calculations included the dual-fuel cycle portion ($q_{cikl}$) and lower heating value ($H_u$), and the chemical composition (C, H, O) was established using the following formula:

$$q_c = \frac{q_{c_D} \cdot H_{L_D} + q_{c_{NG}} \cdot H_{L_{NG}}}{H_u}, \tag{4}$$

where $q_c$ is the overall fuel consumption per cycle (g/cycle); $q_{c_D}$ and $q_{c_{NG}}$ are the diesel and NG fuel consumptions per cycle (g/cycle), respectively; and $H_{L_D}$ and $H_{L_{NG}}$ are the lower heating values of diesel and *NG* (MJ/kg), respectively.

$H_U$, the lower heating value of the fuel (MJ/kg), was calculated by the Mendeleev equation as follows:

$$H_u = 337.5 \cdot C + 1025 \cdot H - 108.3 \cdot O, \tag{5}$$

where:

$$C = C_D \cdot (100 - CCR\ NG) + C_{NG} \cdot CCR\ NG, \tag{6}$$

$$H = H_D \cdot (100 - CCR\ NG) + H_{NG} \cdot CCR\ NG, \tag{7}$$

$$O = O_D \cdot (100 - CCR\ NG) + O_{NG} \cdot CCR\ NG, \tag{8}$$

$$CCR\ NG = \frac{q_{c_{NG}} \cdot H_{L_{NG}}}{q_{c_{NG}} \cdot H_{L_{NG}} + q_{c_D} \cdot H_{L_D}} \cdot 100\%. \tag{9}$$

Here, *CCR NG* is given in percentage, and $C_D$ and $C_{NG}$ are the carbon compositions in diesel and *NG*, respectively.

Energy efficiency parameters:

The indicated efficiency coefficient ($\eta_i$) could be established using the following formula:

$$\eta_i = \frac{3.6 \cdot P_i}{H_{LD} \cdot G_{fD} + H_{LNG} \cdot G_{fNG}}, \tag{10}$$

where $H_{LD}$ *and* $H_{LNG}$ are the lower heat values of diesel fuel and natural gas (MJ/kg), respectively, and $G_{fD}$ and $G_{fNG}$ are the diesel fuel and *NG* consumption (kg/h), respectively.

The effective efficiency could be established using the classical expression $\eta_e = \eta_i \cdot \eta_m$, where $\eta_m$ is the mechanical resistance coefficient, established using the combined mechanical engine losses $P_m$, which can be determined from the experimental indicator diagrams.

## 3. Parametric Analysis of the Combustion Cycle

The results of experimental studies conducted by the authors [43] revealed that ensuring high energy efficiency is one of the most important tasks when the engine is converted to *NG*.

The increase in the share of NG in the dual-fuel D-NG engine results in a decrease in $\eta_i$, particularly when the engine is running at partial load from 0.55 to 0.29–0.33. The corresponding increase in 40–47% of fuel consumption has a negative effect on the engine's toxic components and on the emissions of $CO_2$, which is a GHG. It should be noted that the specific $CO_2$ emissions decreased from 123 to 72 g/kWh (depending on $\varphi_{inj}$) when the engine was running in HLM, and when operating at partial load, GHG increased from 298 to 497 g/kWh. The conversion of DE to dual-fuel provides a $CO_2$ reduction effect owing to the lower carbon content of the fuel compared to diesel. Thus, when engines are converted to gas, the task of improving fuel economy is linked to the complex improvement of key performance indicators.

### 3.1. Calibration of the Mathematical Model for the Research Engine

The initial stage of the research was the calibration of the engine mathematical model parameters with the engine running on diesel and dual fuel (diesel—natural gas): D60-NG40; D40-NG60; D20-NG80.

The subsection presents the verification fragments of the mathematical model with the engine running D and D20-NG80, taking into account the maximum experimental changes of the parameters (compared to D) at the highest part of NG in the dual fuel. The comparison also includes the cut-off values for the range of the fuel injection advance angle 2 (1)° CA BTDC and 13° CA BTDC.

The calibration of MM with the engine running on D, was performed in load mode BEMP = 6 bar with further comparison of calculation and experimental data BEMP = 8 bar, BEMP = 4 ÷ 2 bar (i.e., at higher and lower up to 25% nominal load).

During the modeling of the uncountable modes, only the cyclic portion of the fuel injection was changed, and while evaluating the influence of the fuel injection advance angle—$\varphi_{inj}$. Engine mathematical model (MM) running on D20-NG80, the calibration involves two stages. During the first stage, when changing the D-calibration MM settings, only the elemental chemical composition of the fuel was changed.

However, due to the discrepancy between the parameters of the heat release characteristic ($m$ and $\varphi_z$) according to the experimental data, the modeling errors of the engine energy indicators reach 30–35%, and in individual cases, 50%. Therefore, in the second stage for MM were used the experimentally determined heat release characteristic parameters.

During this step, a positive result was obtained, in parallel the heat release parameters m and $\varphi_z$ were determined according to the [55] method. The realization of fuel in the operation of the engine does not cause any difficulties and is based on the use of integrated energy indicators of the engine.

The results of the MM calibration check are shown in Table 3 and Figure 2.

An error of 1–2% between the calculation and the experimentally determined basic parameters testifies to the accurate calibration of the MM for the tested engine according to all aspects determining the modeling accuracy (especially calculation of heat release characteristics, heat exchanger, etc.). The good agreement between the modeled and experimental indicator diagrams (see Figure 2) also testifies to the correct determination and calculation of the heat release characteristics in MM. Charge air pressure parameters deviation is observed in the experiment low load mode BEMP = 2 bar. The deviations are mainly determined by the modeling of the boost air pressure according to the generalized characteristics of the compressor used in the mathematical algorithm. The second reason is the possible deviation of the heat exchange in the engine cylinder simulation results from the actual data during the combustion cycle. The decision is based on an overall research strategy—the evaluation of pre-controlled engines to convert to dual-fuel operation in the absence of detailed engine technology data including turbocharger characteristics. However, in order to determine the parameters of the heat release with G. Woschni's model testing with the engine running on dual fuel, the necessary corrections of the compressor characteristics were done in further studies (see Tables 3 and 4). Results of the engine parameters that are based on the actual characteristics of the turbocharger and adjusting the

heat exchange in the cylinder according to experimental data of the engine heat balance are presented in Table 3 for verification. As the increase in the excess air ratio shows, it ensures better pressure adequacy in the indicator diagrams (see Figures 2 and 3) by increasing the values of the excess air ratio and the corresponding COP. The increase of the parameter $\eta$ can be explained in turn by a decrease in the duration of fuel combustion (MM heat release) with an increase in $\alpha$ [41,53,57] and a shift in the combustion cycle towards TDC. On the other hand, the adjustment of the heat exchange analytical dependences of empirical coefficients based on of the engine heat balance experiment data also has an influence. The error of $\eta_i$ is ~4% and considered to be an acceptable result of the practical tasks.

**Table 3.** Data of MM calibration check experiment/MM results ($n = 2000$ rpm).

| $P_e$ kW | $p_{me}$ bar | $\varphi_{inj}$ C BTDC | $p_\kappa$ bar | $T_\kappa$ K | $i$ °CA | $p_c$ bar | $p_{max}$ bar | $T_g$ K | $G_{air}$ kg/h | $G_f$ kg/h | $\alpha$ | $\lambda$ | $\eta_i$ | $\eta_e$ | $g_{cycl}$ g/cycl |
|---|---|---|---|---|---|---|---|---|---|---|---|---|---|---|---|
| 6.3 | 2.0 / 1.95 | 1.0 | 1.15 / 1.15 | 346 / 352 | 11 / 9 | 62.2 / 60.8 | 52.2 / 54.1 | 535 / 541 | 133 / 127 | 2.15 / 2.15 | 5.38 / 4.92 | 0.84 / 0.88 | 0.545 / 0.505 | 0.245 / 0.241 | 0.00897 |
| 6.3 | 2.0 / 1.97 | 13 | 1.13 / 1.13 | 345 / 350 | 10 / 8.4 | 61.8 / 61.4 | 78.0 / 76.3 | 524 / 530 | 128 / 124 | 2.10 / 2.10 | 5.22 / 4.87 | 1.3 / 1.23 | 0.559 / 0.530 | 0.256 / 0.251 | 0.00876 |
| 12.6 | 4.0 / 3.9 | 2 | 1.32 / 1.29 | 355 / 353 | 9.5 / 7.1 | 64 / 65.7 | 62 / 62 | 646 / 642 | 136.5 / 129 | 3.34 / 3.34 | 2.88 / 2.82 | 0.94 / 0.95 | 0.455 / 0.462 | 0.316 / 0.310 | 0.0139 |
| 18.9 | 6.0 / 6.0 | 2 | 1.46 / 1.42 | 358 / 354 | 7.0 / 6.6 | 70 / 71.5 | 70 / 71.5 | 724 / 722 | 146 / 140 | 4.61 / 4.61 | 2.19 / 2.21 | 1.0 / 1.0 | 0.452 / 0.456 | 0.344 / 0.346 | 0.0192 |
| 25.1 | 7.94 / 8.1 | 2.6 | 1.58 / 1.55 | 356 / 356 | 6.0 / 6.0 | 77 / 78 | 79 / 80 | 780 / 796 | 158 / 152 | 5.93 / 5.93 | 1.81 / 1.87 | 1.03 / 1.026 | 0.451 / 0.448 | 0.356 / 0.363 | 0.0247 |
| *After corrections of the compressor characteristics* | | | | | | | | | | | | | | | |
| 6.3 | 2.0 / 2.02 | 1.0 | 1.19 / 1.21 | 346 / 352 | 11 / 9 | 62.2 / 62.9 | 52.2 / 54.5 | 535 / 528 | 133 / 135 | 2.15 / 2.15 | 5.38 / 5.23 | 0.84 / 0.87 | 0.545 / 0.522 | 0.245 / 0.249 | 0.00897 |
| 6.3 | 2.0 / 2.03 | 13.0 | 1.17 / 1.18 | 345 / 350 | 10 / 8.4 | 61.8 / 61.7 | 78 / 77 | 524 / 517 | 128 / 133 | 2.10 / 2.10 | 5.22 / 5.13 | 1.3 / 1.25 | 0.559 / 0.549 | 0.256 / 0.259 | 0.00876 |

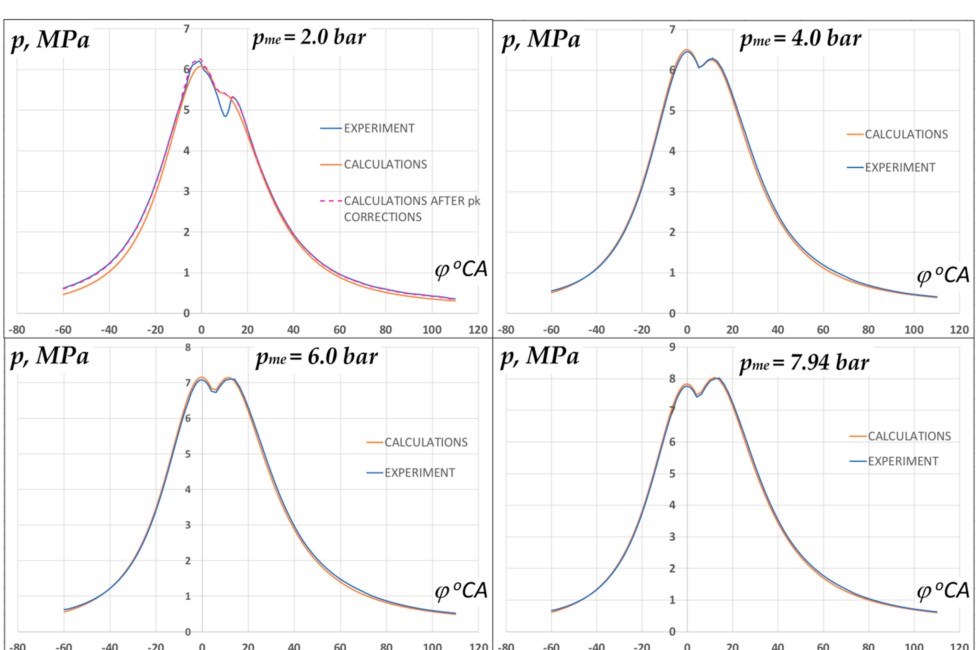

**Figure 2.** Engine combustion cycle mathematical model validation check results ($\varphi_{inj} = 2°$ CA BTDC, $n = 2000$ rpm, diesel).

**Table 4.** Mathematical modeling of the engine combustion cycle using revised parameters $m$ and $\varphi_z$ ($n$ = 2000 rpm).

| | $p_{me}$ = 6 bar | | $p_{me}$ = 4 bar | | $p_{me}$ = 2 bar | |
|---|---|---|---|---|---|---|
| | **Exp.** | **MM** | **Exp.** | **MM** | **Exp.** | **MM** |
| $q_{cikl}$, $g/cikl$ | 0.0169 | 0.0169 | 0.0143 | 0.0143 | 0.01255 | 0.01255 |
| $p_k$, $bar$ | 1.35 | 1.35 | 1.25 | 1.25 | 1.175 | 1.175 |
| $T_k$, $K$ | 333 | 333 | 318 | 318 | 315 | 315 |
| $p_{max}$, $bar$ | 100.6 | 100.0 | 79.0 | 76 | 67.3 | 66.3 |
| $\eta_e$ | 0.345 | 0.345 | 0.273 | 0.276 | 0.155 | 0.157 |
| $\eta_i$ | 0.475 | 0.473 | 0.433 | 0.438 | 0.330 | 0.333 |
| $\eta_m$ | 0.726 | 0.730 | 0.630 | 0.630 | 0.470 | 0.470 |
| $p_{me}$, $bar$ | 5.95 | 5.96 | 3.97 | 4.05 | 1.98 | 2.00 |
| $p_{mi}$, $bar$ | 8.20 | 8.17 | 6.30 | 6.40 | 4.20 | 4.28 |
| $\alpha_\Sigma$ | 2.54 | 2.43 | 2.95 | 2.79 | 3.19 | 2.71 |

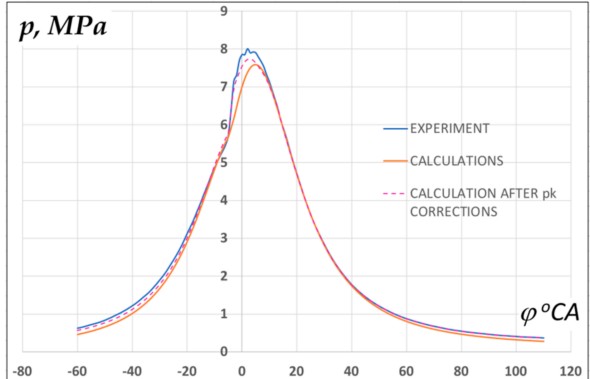

**Figure 3.** Engine combustion cycle mathematical model validation check results ($n$ = 2000 rpm, *BEMP* = 2 bar, $\varphi_{inj}$ = 13° CA BTDC, diesel).

Validation of a mathematical model for practical tasks has been approved in studies at constant engine speed. However, the mathematical model used for a wide range of different types of engine operation also confirms its validity [47–49]. In order to ensure the main adequacy of the simulation modeling results, it is necessary to match the parameters of the charging air unit, combustion cycle heat exchange, and the mechanical COP (friction loss) of the engine with the experimental data. Accurate calibration of heat release and heat exchange dependences of the mathematical model ensures sufficiently accurate modeling results of engine energy indicators (3–4% error). However, it obviously requires approbation and approval of converting the engine to dual-fuel (D-NG).

MM calibration was performed at $\varphi_{inj}$ = 2° CA BTDC. During the verification, the modeling results were evaluated after changing $\varphi_{inj}$ to 13° CA BTDC. The value BEMP = 6 bar deviated the most from the calibration mode. A comparison of the low-load indicator diagrams (see Figure 3) also depicts good coincidence results with the experiment.

It is stated that the MM used in the research adequately reveals the changes of the parameters of the engine running on diesel over a wide load range with the changes of the control parameters. In parallel, the deviation of the $p_K$ and $p_{max}$ parameters from the experiment was observed. According to the algorithm of the mathematical model, one of the main factors influencing the simulated heat release characteristic in partial load modes is the coefficient of excess air. Its decrease compared to the experiment became the main reason for the increase of heat release characteristic in the duration $\varphi_z$ and the decrease in $p_{max}$, respectively (see Figure 3). In the modeling of dual-fuel engine parameters, this aspect was taken into account to ensure $p_{max}$ and $\alpha$ consistency with the experimental data. Based on the research results, the necessary condition for the parametric analysis of the used [46] mathematical model is the modeling of the parameters of the charge air unit,

based on the MAP (MAPs of VNT/VGT) data of its characteristics. As mentioned, the calibration of the MM with the engine running on diesel showed negative results in the case of dual fuel use. To determine that the MM discrepancy is caused by a significant deviation of the heat release (HRR) characteristics from the actual experimentally determined data. The progress of the engine differential experimental values HRR $\left(\frac{dQ}{d\varphi}\right)$ in transferring the engine to operation from D to D-NG fuel is well shown in Figure 4a–c.

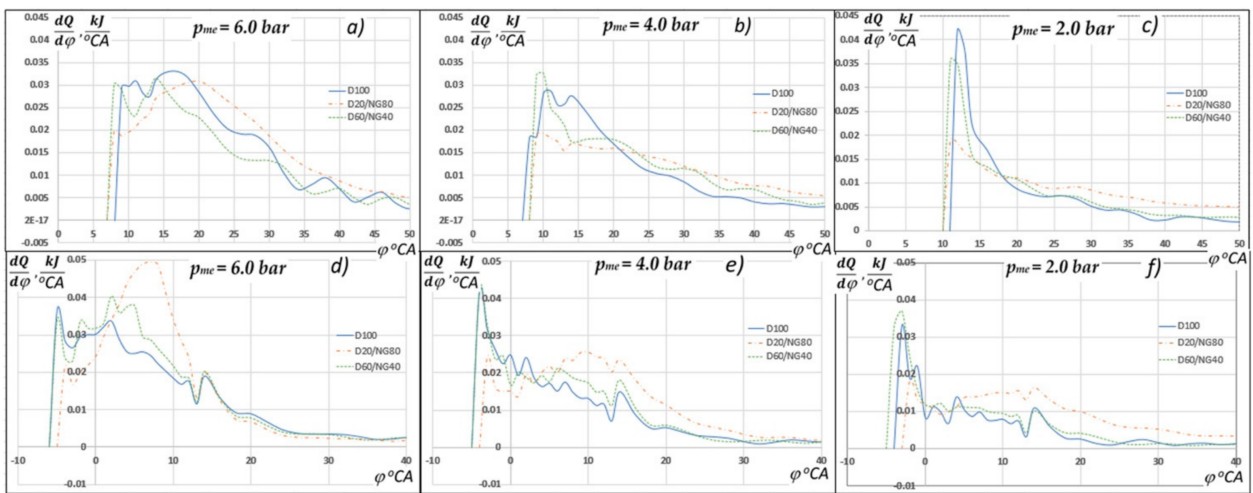

**Figure 4.** Changes in heat release characteristic of engine when transfer from diesel to natural gas fuel where *(n = 2000 rpm)*: (**a**–**c**) is $\varphi_{inj}$ = 1° CA BTDC and (**d**–**f**) *is* $\varphi_{inj}$ = 13° CA BTDC.

As the proportion of NG in the dual fuel increases, strong changes HRR take place: the first kinetic combustion phase of the fuel decreases and the diffuse combustion phase extends into the expansion cycle. Particularly significant changes are characterized by a decrease in engine load and an increase in the proportion of NG. In dual fuels, for example, (Figure 4a) at BEMP = 6 bar, the maximum value of HRR in the kinetic phase decreases from 0.03 $\frac{kJ}{deg}$ to 0.02 $\frac{kJ}{CA^o}$ from D to D20-NG80; in low load mode $p_{me}$ = 2 bar the corresponding changes are from 0.042 $\frac{kJ}{deg}$ to 0.019 $\frac{kJ}{deg}$. Due to decrease in the kinematic component, the heat release in the second diffusion phase increased, especially in $\varphi_{inj}$ = 13° CA BTDC: correspondingly maximums values increased from 0.035 $\frac{kJ}{deg}$ to 0.05 $\frac{kJ}{deg}$. The trends of HRR change do not change qualitatively in the studied $\varphi_{inj}$ range 1 ÷ 13° CA BTDC (see Figure 4d–f). In terms of increasing energy efficiency, the increase in heat release intensity in the diffusion phase is positive, but is associated with a parallel reduction in total combustion time. The characteristic of the HRR change according to $\varphi$ °CA suggests that the heat release characteristic could be considered as single-phase, the kinetic and diffuse phase separation is more conditional.

HRR data confirmed by authors comparative studies of the heat balance with engine running on D and NG [45]. When the engine is running on NG, the prolonged combustion process in the expansion stroke results in a significantly increased heat transfer to the cooling system, which contributes to deterioration of energy efficiency. In LLM modes, with the maximum tested NG fraction in dual fuel 80%, the heat loss to the engine cooling system increases up to 2.5–3.0 times. At that time, the indicative COP cycle decreases from 0.55 to 0.33. The experimental data of the heat balance confirm the determined parameters of the combustion process. The observed feature became an additional justification for the use of single—phase I.Vibe MM. In the investigated variant in the absence of the targeted engine design for D-NG fuel, the combustion period becomes very prolonged. If in high load modes (BEMP = 6 bar) the relative burning time according to I.Vibe model increases by ~5° CA, but in medium (BEMP = 4 bar) and low (BEMP = 2 bar) load modes $\varphi_z$ compared to D variant lasts from 44° CA and 35° CA to 110° CA and 200° CA, respectively. At that time,

the values of $\varphi_z$ calculated by G. Woschini [53] decrease as it became the main reason for inaccurate modeling. It is obvious that in order to achieve the versatility of the parametric analysis method while engine working on a wide range of fuels (including D-NG fuels), it is necessary to refine the phenomenological model of heat release characteristics or to apply a more detailed model of physical processes in the engine cylinder.

In the performed research stage, in order to confirm the rationality of the application of the parametric analysis method, it is necessary to calibrate the MM separately for the D-NG variant, based on the real experimental data. The results of modeling of engine parameters when calibrating MM based on experimental data of D-NG variant are presented in Table 4.

The decision to calibrate the engine MM with the engine running on D-NG fuel is confirmed by a sufficiently low error between the simulation and the experimental results, error is between 3–4%, it is important to note that error is low—in the whole range of known $\varphi_{inj}$ changes.

A good agreement between the results of the experiment and the modeling is substantiated by the comparison of the indicator diagrams of the relative form of the heat release characteristics, the fragments of which are presented in Figure 5. The maximum error does not exceed 5–7%.

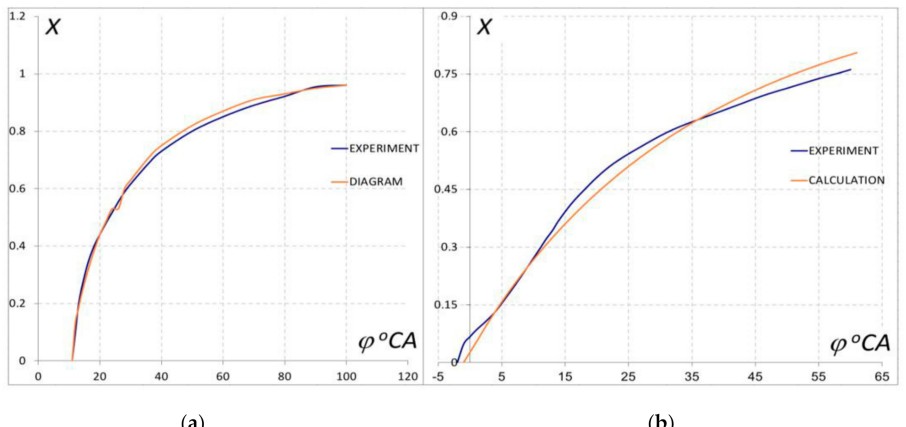

(a)          (b)

**Figure 5.** Engine heat release characteristics working with dual fuel D20-NG80 (*n* = 2000 rpm, BEMP = 2 bar): (**a**) -$\varphi_{inj}$ = 1° CA; (**b**) -$\varphi_{inj}$ = 13° CA.

Based on the results of MM calibration, the ways and limitations of its rational use in parametric analysis were identified, as well as directions for improvement.

### 3.2. Analysis of the Combustion Cycle Factors That Determine $\eta_i$

Methods for improving energy performance, $\eta_e$, $\eta_i$, through parametric analysis have been extensively studied, particularly in the 1980s, with the onset of the energy crisis [49]. Researchers attributed the improvement of $\eta_i$ to the degree of compression ($\varepsilon$), excess air coefficient ($\alpha$), process dynamics (degree of pressure increase ($\lambda$), ($p_{max}$) maximum cycle pressure, air pressure after compression ($p_k$), ambient air ($T_{at}$), charge air temperature ($T_k$)), and other secondary factors [37,38]. Studies have been performed to determine the analytical dependences in the state of linear equations $\eta_i = (\varepsilon, \alpha, \lambda, p_k, T_k)$ for a certain class of DEs [58]. The constants of the linear equations that determine the accuracy of the calculations in most cases were determined from statistical experimental data and by providing analytical or graphical formulas for the dependencies of the operational parameters. Analogous solutions have been implemented for DEs of V. Yuzhin ships [38]. The main disadvantage of this method is the need to update the information base constantly with the values of the factors that influence energy indicators [36,39].

Several studies on the characteristics of the physical process—heat release—occurring within the cylinder and determining the COP of the combustion cycle and, largely, the energy efficiency have been conducted [40,41,57,59]. The methods developed in these

studies and the statistical methods for processing experimental data have served as foundations for identifying the influence of heat release duration and variation in the value of the COP of the combustion cycle [60–62]. Furthermore, to substantiate this relationship analytically for analysis of the cycle of the dual-fuel engine, Stechkin's study should be used as a reference [59]. Stechkin [59] investigated the relationship between the theoretical cycle, $\eta_{t_0}$, and the combustion process rate and provided a series of analytical solutions (Figure 6).

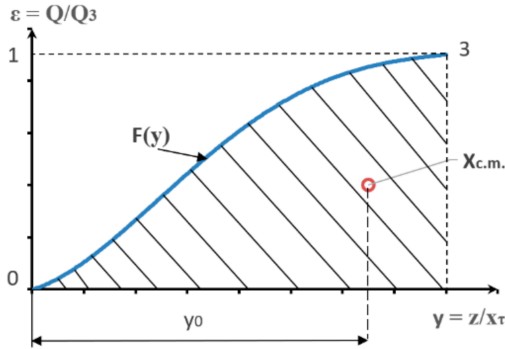

**Figure 6.** Parameters of the heat release law.

($Q_3$: heat release in the cycle; $y_0$ field center abscissa; $F(y)$ heat release curve; $x_\tau$ heat release duration).

The analytical solutions provided can be compared based on their nature, thereby enabling evaluation of the COP for the rapid combustion cycle in accordance with the analytical equations of the law of thermodynamics (11):

$$(\eta_{t_0} - \eta_t)\cdot\varepsilon^{k-1} = (k-1)\cdot\frac{x_\tau^2}{v_0}\cdot2\cdot(1-y_0)\cdot(2-2y_0-\Delta) = (k-1)\cdot\frac{x_\tau^2}{v_0}\cdot2\cdot\Omega. \quad (11)$$

The difference between the theoretical cycle with finite combustion rate, $\eta_t$, and the rapid combustion cycle efficiency ratio $\left(1 - \frac{1}{\varepsilon^{k-1}}\right)$ is determined by three parameters of the heat release law: $x_\tau$, $\Delta$, and $y_0$ ($\Delta$: area under the curve of heat release). The final $\eta_t$ correction $(\eta_{t_0} - \eta_t)$ depends on the combustion time $(x_\tau^2)$, expressed in parts of the CC volume, $v_0$, and parameter $\Omega$ is the heat release law factor in studies related to the heat release characteristics of the area under the curve of center mass location. As a result of a series of simplifications, the following final expression is generated: $(\eta_{t_0} - \eta_t)\cdot\varepsilon^{k-1} = \frac{x_\tau^2}{v_0}\cdot2\cdot\Omega$. Under $\Omega = invar$, $(\eta_{t_0} - \eta_t)$ correction for the selected $\varepsilon$ is determined only by the heat release duration, $\chi_\tau$. This statement has been used as one of the bases for the analysis of multivariate MM results.

In modern DEs, with a relatively short $x_\tau$, the correction is not significant. However, the more than four-fold increase in $x_\tau$ noticed during the experimental study on the DE has a significant effect on $(\eta_{t_0} - \eta_t)$ and $\eta_i$. The finite element method has been used for Equations (12) and (13) [63]: $x_c = \frac{\sum x_i m_i}{\sum m_i}$; $y_c = \frac{\sum y_i m_i}{\sum m_i}$, or the shape of the proposed finite elements in the expression of the equation; the product of the first and second factors of the additive sum is equal to the area of the elementary element, $(\varphi_{i-1} - \varphi_i)$, from the third factor to the ordinate of the centers of mass of the elements $(\varphi_{i-1} - \varphi_i)$. In the structural expression of Equation (12), the third factor of the product equals the centers of mass of the elements $(\varphi_{i-1} - \varphi_i)$:

$$X_{\text{center of mass ordinate}} =$$
$$= \left[\left(\frac{x_0+x_1}{2}\right)\cdot(\varphi_1 - \varphi_0)\cdot\frac{(x_0+x_1)}{2\cdot2} + \left(\frac{x_1+x_2}{2}\right)\cdot(\varphi_2 - \varphi_1)\cdot\frac{(x_1+x_2)}{2\cdot2} + \cdots\right]$$
$$\div \left[\left(\frac{x_0+x_1}{2}\right)\cdot(\varphi_1 - \varphi_0) + \left(\frac{x_1+x_2}{2}\right)\cdot(\varphi_2 - \varphi_1) + \cdots\right], \quad (12)$$

$$y_{\text{center of mass abscissa}} =$$

$$= \left[ \left( \frac{x_0 + x_1}{2} \right) \cdot (\varphi_1 - \varphi_0) \cdot \left( \varphi_0 + \frac{(\varphi_1 - \varphi_0)}{2} \right) + \frac{x_2 + x_1}{2} \cdot (\varphi_2 - \varphi_1) \cdot \left( \varphi_1 + \frac{(\varphi_2 - \varphi_1)}{2} \right) \right]$$
$$\div \left[ \left( \frac{x_0 + x_1}{2} \right) \cdot (\varphi_1 - \varphi_0) + \left( \frac{x_1 + x_2}{2} \right) \cdot (\varphi_2 - \varphi_1) \cdots \right]. \tag{13}$$

Fragments of the results of the study on the dynamics of the heat release characteristics at different engine loads with dual-fuel D-NG during the center mass change cycle are presented in Figure 7.

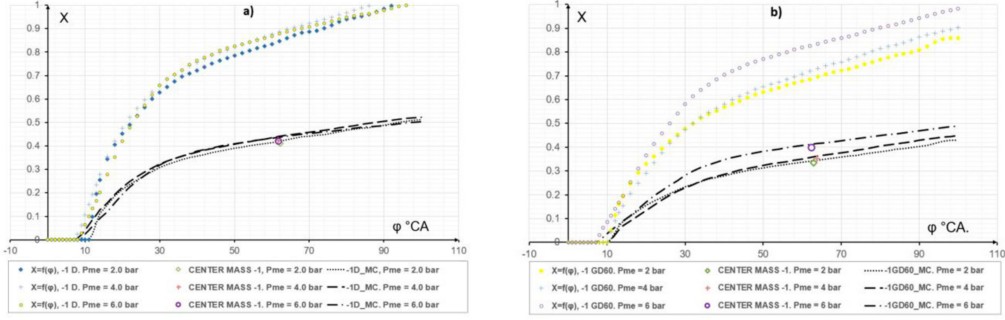

**Figure 7.** Dynamics of change of coordinates of mass centers of heat release characteristics ($n$ = 2000 rpm)$:x = f(\varphi)$: ○, +, ◊ - $\varphi$ = 1° CA BTDC, (**a**) BEMP = 2 bar, (**b**) BEMP = 4 bar.

The implemented analytical solutions shown in Figure 7 demonstrate the dynamics of the change in the center of mass of the heat release characteristic during the cycle. Essentially, the dynamics of change in the abscissa ($\varphi_c$ according to Stechkin [54]) of the centers of mass for modes BEMP = 2, 4, and 6 bar and the final value of $\varphi_c$ remain constant for all variants examined from D to D20-NG80, with minor exclusions. Analysis of the diagrams indicates that with the engine operating in the dual-fuel mode, where NG is responsible for a significant share of energy, increasing $\varphi_{inj}$ within the range analyzed, and possibly within a wider range, may become an effective tool for increasing the energy cycle.

Advancing the diesel fuel injection phase, $\varphi_{inj}$, significantly increases the heat release dynamics in the expansion cycle when $p_{max}$ and $T_{max}$ are reached. It was found that the ordinate values of the center of mass, and accordingly the current values of $X_i$, are higher when the engine is running on dual fuel compared to running on diesel. When the engine is running on diesel, the increase in heat release dynamics at higher $\varphi_{inj}$ values is characteristic of the initial heat release headers. In the range of 25–40° CA after TDC, the values of $X_i$ become equal, and the ordinates of the values of the higher center of mass ($X_i$) continue to prevail at lower $\varphi_{inj}$. Thus, when the engine is running on diesel, it is reasonable to expect a lower $\eta_i$, leading to a further decrease in the combustion cycle dynamics.

Thus, the generated results support the application of the method of parametric analysis of the combustion cycle in association with the indicators of the heat release characteristic. In contrast, based on the theoretical analysis, Equation (14) [59] describes the relationship between the engine cycle, $\eta_e$, and heat release characteristics as a function of $x_\tau$ duration and shape parameters:

$$(\eta_{t_0} - \eta_t) = (k-1) \cdot \frac{x_\tau^2}{v_0} \cdot 2 \cdot \Omega. \tag{14}$$

A similar conclusion was reached in the study by Lebedev et al. [47], in which the main factors influencing the engine duty cycle efficiency factor $\eta_i$ were based on the change in fuel burn duration and $p_{max}$ in the cylinder. Thus, the value of $p_{max}$, under the condition $p_{mi}$, $\alpha$, is described by the amount of heat when $Q$ ($p_{max}$) is reached while $\varphi_{p_{max}}$° CA. Considering this, the value of $Q$ ($p_{max}$) at the optimal start phase of combustion depends

directly on the dynamics of the combustion process or, in other words, on the shape of the heat release characteristic.

Based on this, the value of $p_{max}/p_k$ [51] in this work represents the influence of the shape of the law of heat release on the value of $\eta_i$. In contrast, the $p_{max}/p_k$ ratio is the product of $\varepsilon^k \lambda$, the two other indicators that determine the level of $\eta_i$, together with $\alpha$ [56,64]. According to statistics summarising a number of experimental studies [52,53], the influence of the excess air coefficient, $\alpha$, on $\eta_i$ is expressed as an analytical dependence (15):

$$\varphi_z = \varphi_{z_0} \cdot \left(\frac{\alpha_0}{\alpha}\right)^{0.5} \cdot \left(\frac{n}{n_0}\right)^{0.6}, \tag{15}$$

where $\varphi_z(\varphi_{z_0})$ is the heat release time according to the I. Vibe model; the value $(\varphi_{z_0})$ characterizes the duration of heat release during operation of the motor in the basic numerical mode, which is usually accepted as the nominal power mode. The equation shows $\varphi_{z_0}$, which also indicates the dependence of $\eta_i$ change on the $\alpha$ and $n$ parameters in the presence of the form factor $m$ [51] and $\Omega$ [59]; otherwise, the mass of the center of the heat release characteristic is the same.

### 3.3. Optimization of DE Combustion Cycle Parameters

In the works by Ivanchenko [42,64], the MM of the relationship between $\eta_i$ and characteristic indicators of the combustion cycle were applied to highly supercharged DEs, forced according to $p_{mi}$. Practical application of the model essentially consists of a numerical multivariate experiment that serves as the basis for developing the generalized graphic dependences of the combustion cycle parameters. The influence of the $p_{max}$ limitation of the cycle on the fuel cost efficiency and engine heat stresses, described by $\eta_i$ and $\alpha$, respectively, is assessed. Simultaneously, the selection of the rational combination of cycle indicators to achieve the anticipated value of the engine energy efficiency $\eta_i$ is addressed.

The method is characterized by the convenience of its practical application; however, one of the limitations is the set of accepted fixed indicators of the heat release law: the heat release duration and heat release shape parameters. In fact, the variation in the forcing rate according to $p_{mi}$ determines significant changes in this important indicator in relation to the energy efficiency of the cycle. Moreover, it is not only $p_{me}$ but also the combination of the combustion cycle parameters ($\varepsilon$, $\lambda$, $\alpha$, etc.) that have a fairly large influence on the heat release duration and shape.

Variational modeling of the duty cycle was performed at different values of $\varepsilon_0$, $\alpha$, and $p_{max}/p_k$. Based on the MM results, generalized graphical dependences were obtained that represented the relationships between the indicated efficiency factor, complex forcing according to the $p_{mi}$ parameter, $(p_{max}/p_{mi}) \cdot (350/T_k)$, and $\alpha$, $p_{max}/p_k$ parameters. The obtained generalized graphical dependences allow to evaluate the methods of rational in-cylinder engine process execution ($\varepsilon$, $p\, T_k$, $\alpha$, $\varphi_{inj}$) for the specific structural configuration engine model in order to achieve energy efficiency under acceptable constraints.

This method is easy to apply, however, one of its provisions limits its practical application to convertible operation for dual-fuel engines. The method implementation algorithm adopts fixed heat release law indicators: heat release time and heat release law parameter $\varphi_c$. The burning time, according to the accepted accuracy of 0.3%, was determined in the form of a constant equal to 8 $\varphi_c$, and $\varphi_c$ was assumed to be $12.4 \pm 4°$ CA BTDC.

The application of this parametric analysis method to dual-fuel engines in relation to the significant changes in the parameters of the heat release characteristic should be supported by analytical, experimental, and MM research development. Indeed, the change in the forcing level, according to $p_{mi}$, leads to the perceived changes in this important indicator for the energy efficiency of the cycle. Experimental studies have also demonstrated a significant increase in burning time when the engine is running on dual fuel, particularly in the LLM [43]. Thus, considering the significant influence of the burning time, the use of real values of the heat release time in the studies became an essential change in the method

used in the parametric analysis. The method used in parametric analysis of the combustion cycle was approved through multivariate numerical research using the IMPULS software.

### 3.4. Investigations of the Optimization of Dual-Fuel D-NG Engine Combustion Cycle Parameters

We investigated the ranges of change of the initial data: $\alpha = 1.7$–$4.0$; $\varepsilon = 15.5$–$23.5$; $\varphi_{in\,j} = 1$–$35°$ CA BTDC (differentiated with different $p_{mi}$ values). Figures 4 and 5 present the results of the parametric analysis of the operating process at the limit values of $p_{mi} = 4.2$ bar and $p_{mi} = 8.2$ bar in the load range with the engine running on diesel fuel and dual fuel, respectively. The sufficient accuracy of the MM results for solving practical problems is confirmed by their comparison with the experimental data. The results of the experiment in the diagrams are presented in the form of local points in the coordinates of $\alpha$–$p_{max}/p_k$. The differences in $\eta_i$ do not exceed 2–4%.

We have received the traditional form of relations between the $\eta_i = f(p_{max}/p_k, \alpha, \varepsilon = invar.)$ and the motor forcing parameter, $\Pi = f(p_{max}/p_k, \alpha, \varepsilon = invar.)$. The curves of $\eta_i = f(pP_{max}/p_k, \alpha, \varepsilon = invar.)$ for different $\varepsilon$ are given by the lines of rounded shapes. The obtained data show that at $p_{mi} = 8.2$ bar ($n = 2000$ rpm) and $\alpha = 2.3$, $\eta_i$ is close to its maximum achievable value at point 2. Increasing $p_{max}$ above 9 MPa (lines 1–2–3) has virtually no effect on $\eta_i$. In addition, the increase of reserve $\eta_i$ is also used because of the increase in $\varepsilon$ above the limit of 19.5 designed for the engine. Simultaneously, $p_{max}/p_k$ increases above 74.5 pcs., and the increase in $\alpha$ has no effect on $\eta_i$. An analogous conclusion can be reached by evaluating the LLM $p_{mi} = 4.2$ bar (Figure 4). Increasing $\alpha$ above 4.5 units also does not affect the value of $\eta_i$. An increase in $\varepsilon$ and an extension of the $p_{max}$ limit (shift of line 1–2–3–4 to the 5–6–7–8 position) lead to a decrease in the value of $\eta_i$ (shift of line 1–4 to the 5–8 position). The results of the parametric analysis indicate that in the case of HLM, when the motor operation process execution is achieved and the $p_{max}$ limits are set, the achieved level of $\eta_i$ is close to the optimal one. It is known that radical engine improvements, such as the installation of a storage fuel injection system, would quantitatively change the relationships shown in Figure 8 and expand the potential for energy efficiency improvements.

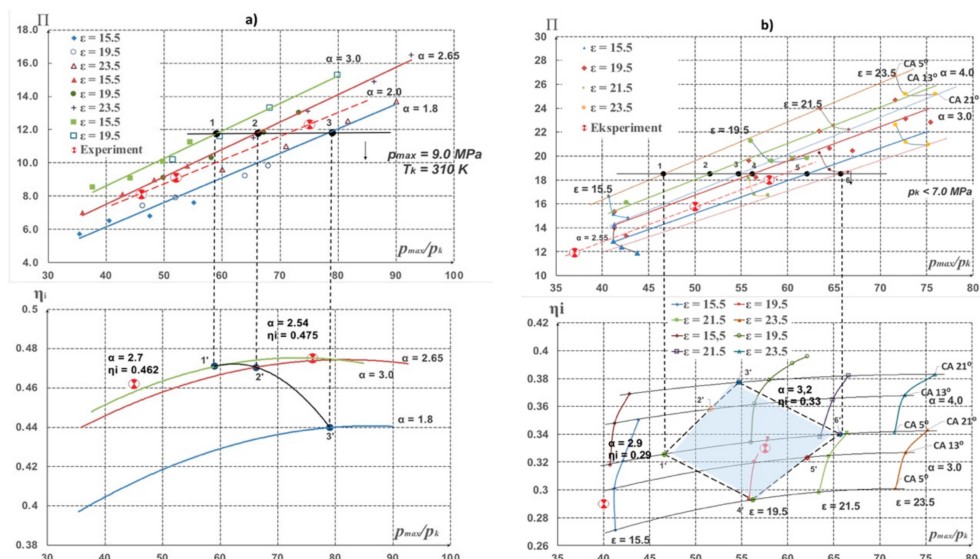

**Figure 8.** Results of parametric analysis of the combustion cycle with the engine running ($n = 2000$ rpm) on dual-fuel (**a**) D20-NG80 at $p_{mi} = 8.2$ and (**b**) D20-NG80 fuel at $p_{mi} = 4.2$ bar.

Increasing $\varepsilon$ under $\varphi_{inj} = invar.$ positively influences $\eta_i$ throughout the range analyzed, which is $\varepsilon = 15.5$–$23.5$ units. However, the value implemented in the engine, which is $\varepsilon = 19.5$, corresponds to the optimum value of $\eta_i$. In summary, from the analysis of the change of nomogram $\Pi$, $\eta_i = f(p_{max}/p_k, \alpha = invar.)$, the following conclusions are drawn.

- Considerable reduction in the heat release dynamics under the dual-fuel D-NG LLM operation of the engine strengthens the influence of the change in $\varphi_{inj}$ on $\eta_i$ within a wide range of $\varepsilon$ and $\alpha$, and, in parallel, reduces the influence of $\varphi_{inj}$ on $p_{max}/p_k$ or $p_{max}$.
- The range of change of $\varphi_{inj}$ may be expanded with the aim of improving $\eta_i$ without posing a risk of reaching the limit values of the reliability indicator, $p_{max}$.

The numerical analyses reveal a strong influence of the fuel injection timing phase, $\varphi_{nj}$, on $\eta_i$, and also have good correlation with the experimental research results. Irrespective of the load, the advancement of the injection timing, $\varphi_{inj}$, within the range analyzed, which is $\varphi_{inj}$ = 1 to 13° CA BTDC, influences the increase in $\eta_i$ at different levels under diesel-fuel versus dual-fuel operation of the engine. The increase in the NG portion in the dual fuel from 0 to 80% leads to different effects of the change in $\varphi_{inj}$: 1–18% under $p_{mi}$ = 4.2 bar, 4–21% under $p_{mi}$ = 6.2 bar, and 7–17.5% under $p_{mi}$= 8.2 bar load modes (Table 5).

**Table 5.** Influence of $\varphi_{inj}$ phase advance on $\Delta\eta_i$ increase, in the range from 1 to 13° CA BTDC ($n$ = 2000 rpm).

| | $p_{mi}$ = 4.2 bar | | | | $p_{mi}$ = 6.2 bar | | | | $p_{mi}$ = 8.2 bar | | |
| | **D** | **NG40** | **NG60** | **NG80** | **D** | **NG40** | **NG60** | **NG80** | **D** | **NG40** | **NG60** | **NG80** |
|---|---|---|---|---|---|---|---|---|---|---|---|---|
| $\Delta\eta_i$ % | 1 | 11 | 16 | 18 | 4 | 11 | 15 | 21 | 7 | 9 | 13 | 17.5 |

Owing to the low intensity of the fuel combustion kinetics and work process under D-NG-fuel engine operation, a relatively high share of heat is released at later phases of the expansion stroke. Hence, $\varphi_{inj}$ advancement to 13° CA BTDC led to an intensive increase in $\eta_i$, both during the experiment and during the modeling. Under DE operation, the change in $\varphi_{inj}$ led to a 2% increase in $\eta_i$ (maximum $\eta_i$ achieved at $\varphi_{inj}$ was equal to 7° CA BTDC), whereas under dual-fuel D20-NG80, $\eta_i$ increased by 17%. In the numerical modeling, the advancement of $\varphi_{inj}$ to 30° CA BTDC had a positive influence on the increase in $\eta_i$, but at a lower intensity than that of $\eta_i$ when the range was between 1 and 13° CA BTDC (see Figure 8).

Investigation of the combination of optimum parameters of the engine combustion process under the $p_{mi}$ = 4.2 bar mode essentially does not differ from the aforementioned HLMs; for example, the assessment of the rational combination of $\varepsilon$, $\varphi_{inj}$, and $\alpha$ was performed under the condition of $p_{max} \leq 6.9$ MPa. Under $p_{max}$ = 6.9 MPa, the value of parameter $\Pi$ is $\Pi = \frac{p_{max}}{p_{mi}} \cdot \frac{350}{T_k} = \frac{6.9}{0.42} \cdot \frac{350}{310} = 18.5$. The fields marked by points 1′–3′–6′–4′ graphically define the combinations of $\varepsilon$, $\alpha$, $\varphi_{inj}$ that improve $\eta_i$. Hence, the parametric $\eta_i$ $\frac{p_{max}}{p_{mi}} \cdot \frac{350}{Tk} = f(p_{max}/p_k; \alpha; \varepsilon = idem)$ analysis method has been analytically substantiated and verified through experiment and numerical investigation for the purposes of practical application with the aim to improve the energy efficiency parameters of an engine converted to dual-fuel operation. At the same time, the research showed rational aspects of further evaluation and improvement of the method, with a view to its real practical application. Among other things, attention must be paid to adapting it to the different speeds when assessing the operation of the engine over a wide speed range. It is expected that the qualitative dependence of $\eta_i$ on the main factors of the combustion cycle ($\alpha$, $\varepsilon$, $\mu$, $p_k$, $T_k$) should not change. However, it must be taken into account that the modification of the method was related to the actual use of the heat release characteristic duration $\varphi_z$ to form the nomogram in Figure 8. On the other hand, $\varphi_z$ together with $\alpha$ is determined by the engine speed (for example see Equation (8)). Meaning qualitatively unchanged for each engine speed, it will be necessary to use quantitatively separate nomograms. The partially made assumption is confirmed by the results of the authors' previously studied engines 150/180 mm (2000–1300 rpm) and 165/185 mm (1850–1200 rpm) [51,60]. It is also important to adapt the dependences of heat release and heat exchange from the same basic calibration mode to different dual fuel compositions in G. Woschni's analysis.

## 4. Conclusions

In this research, one of the problems of decarbonization of transport was investigated, by transferring the operating fleets of diesel engines to work on natural gas. The results of the research confirm the hypothesis about the rationality of applying parametric analysis, assessing the energy efficiency potential and determinants of a dual-fuel diesel engine. The conversion of high-speed 79.5/95.5 mm DE) with a conventional fuel injection system for operation with a dual-fuel D-NG and the relationship between the indicator efficiency, combustion cycle performance parameters and characteristics, and rational optimization directions to achieve energy efficiency have been studied. Methodological solutions for the parametric analysis of the combustion cycle have been identified for practical use and applied to the research engine.

It was found that the energy efficiency of a DE for convertible dual-fuel operation, especially in LLMs, is determined by the combustion time of gaseous fuel, which increases the pilot of the fuel injection advance phase $(\varphi_{inj})$, with a simultaneous decrease in the compression ratio. On this basis, the conversion of a converted DE with a conventional fuel system is a rational $\varphi_{inj}$ advance compared to the process by which it is achieved, in practice, in dual-fuel engines with accumulative CR-type systems.

Factors that determine the energy efficiency: heat release characteristics $X = f(\varphi)$, duration $\varphi_z$, center mass abscissa, or form factor $m_z$ are described with indicative process performance parameters $\varepsilon$, $\alpha$, $\lambda$, $p_k$, and $T_k$, form the parametric analysis:

$$\eta_i \frac{p_{max}}{p_{mi}} \cdot \frac{350}{T_k} = f(p_{max}/p_k, \, \alpha). \tag{16}$$

After applying the method of parametric analysis in a wide range of engine operating modes, the reserves of improvement of the $\eta_i$ and the rational directions for their practical implementation are as follows:

- In the HLM ($p_{mi}$ = 8.2 bar), in both diesel and dual-fuel D20/NG80 applications, the design *(ε)* and indicative process performance parameters (*α*, cycle dynamics associated with $\varphi_{inj}$) implemented in the engine are optimal and ensure $\eta_i$ values close to the maximum;
- In the LLM, owing to the reduced heat release dynamics ($\varphi_z$ increases from 60° CA in HLM to 200°), the main factors influencing $\eta_i$ become $\varphi_{inj}$ and $\alpha$, whereas $\varepsilon$ optimization is not expedient;
- In contrast to the HLM, strong correlations between $\eta_i$ and $\varphi_{inj}$ occur over the entire range of $\alpha$ and $\varepsilon$ changes, in addition to increasing the $\varphi_{inj}$ from the static 1° to 30° CA BTDC and more;
- The reserves of $\eta_i$ increase in the LLM by optimizing $\varphi_{inj}$, $\alpha$, and $p_k$, and this increase is estimated at ~10%, maintaining the set value of $p_{max}$, which determines the mechanical reliability limitations.

The application of the method of parametric analysis of DE combustion cycle for conversion to dual-fuel operation is proposed to include numerical research methods of heat release characteristic parameters, formation of a MM of the researched object; variational modeling of the formation of nomogram $\eta_i$: Π = f(p_{max}/p_k; α; \varphi_{inj} = idem), and its parametric analysis to achieve the maximum $\eta_i$ under the established limits of reliability ($p_{max}$, etc.) parameters. For initial energy parameters, potential improvement of specific engine model heat release characteristics identification methods must be used. The proposed methodology is seen as a theoretical tool for a dual-fuel conversion model for in-service engines and has benefit of a practical use of a fast application in the industrial field. The use of a refined mathematical model for optimizing engine parameters, provided for in further research by the authors, is considered as the basis for a real practical application of the methodology.

**Author Contributions:** S.L. Conceptualization; S.L. methodology; S.L. and T.Č. software; T.Č. validation; S.L. Formal analysis; S.L. and T.Č. investigation; T.Č. data curation, S.L. and T.Č. writing—review and editing. All authors have read and agreed to the published version of the manuscript.

**Funding:** This research received no external funding.

**Institutional Review Board Statement:** Not applicable.

**Informed Consent Statement:** Not applicable.

**Data Availability Statement:** Data sharing not applicable.

**Acknowledgments:** The author is grateful to colleagues at VGTU for the warm welcome and the opportunity to use the existing engine testing facilities at the VGTU Transport Engineering Faculty during the experimental research.

**Conflicts of Interest:** The authors declare no conflict of interest as the parametric analysis is authors decision.

## Nomenclature

| | |
|---|---|
| $n$ | Engine speed (rpm) |
| $T_g$ | Exhaust gas temperature ($K$) |
| $T_K$ | Air temperature after compression ($K$) |
| $P_e$ | Brake power (kW) |
| $\alpha$ | Air-access coefficient |
| $\varepsilon$ | Compression ratio |
| $\varphi_{inj}$ | High reaction fuel injection time ($°CA$) |
| $H_U$ | Lower heating value (kJ/kg) |
| $\eta_m$ | Mechanical efficiency |
| $\eta_i$ | Indicated efficiency |
| $\eta_e$ | Effective efficiency |
| $p_{max}$ | Maximum cycle pressure in the cylinder (bar) |
| $p_k$ | Air pressure after compression (bar) |
| $p_c$ | Pressure of compression in the cylinder (bar) |
| $\lambda = \frac{p_{max}}{p_c}$ | Cylinder pressure increase rate |
| $X = f(\varphi)$ | Relative heat release ratio ($°CA$). |
| $\frac{dQ}{d\varphi}$ | Heat release rate (kJ/$deg$) |
| $p_{me}$ | Brake effective mean pressure (bar) |
| $p_{mi}$ | Indicated mean pressure (bar) |
| $CA_{50}$ | Half of heat released during cycle ($°CA$) |
| $T_{max}$ | Maximum combustion temperature |
| $\varphi_z$ | Conditional combustion duration |
| $m$ | Form factor |

## Abbreviations

| | |
|---|---|
| MM | Mathematical model |
| COP | Effective coefficient of performance |
| CR | Common rail |
| CCR NG | Co-combustion ratio of natural gas |
| CC | Combustion chamber |
| DIT | Diesel fuel injection timing |
| HRF | High reaction fuel |
| HRR | heat of releas rate |
| BMEP | Brake effective mean pressure |
| HLM | High load mode |
| LLM | Low load mode |
| TDC | Top dead center |
| BTDC | Before top dead center |
| CA | Crankshaft rotation angle |
| D | Diesel fuel |

LNG    Liquefied natural gas
*PM*    Particulate Matter
NG    Natural gas
$NO_x$    Nitrogen oxides
$CO_2$    Carbon dioxide

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
