# Peer review of "Parametric Analysis of the Combustion Cycle of a Diesel Engine for Operation on Natural Gas"

_sustainability, doi:10.3390/su13052773_

Round 1
Reviewer 1 Report
With reference to the authors Rebuttal:
Comment 2:
Figure 2 at pme = 2 bar and Figure 3 show a very strong misalignment between theoretical and experimental data. The authors should try to work more on this point.
The reviewer is not convinced that the disagreement on pressure traces at low load (in compression and combustion) are due to discrepancies between plenum pressure and excess air. If the authors are sure about it, it would be useful to add a numerical / experimental comparison plot for all points (those for which cycles are shown) both for the plenum pressure and excess air. This would convince both the reviewer and the reader that this modeling difficulty occurs only for that specific low load point.
Comment 7:
The analysis was conducted at an engine speed of 2000 rpm.
The authors insert a brief consideration at the end of paragraph 3.4 simply by stating that the proposed method must be adapted to different rotation regimes. However, it is not discussed how reliable it is on the entire speed range of the engine.
Reviewer 2 Report
The study presented is interesting.
Please see minor errors, namely in line 372 and the numbers of the figures referred to in the body of the text with the figures to which, in fact, the authors want to refer.
Author Response
Comment of the Reviewer
“Please see minor errors, namely in line 372 and the numbers of the figures referred to in the body of the text with the figures to which, in fact, the authors want to refer.“
Dear Sir / Madam,
Thank you very much for your valuable observations, comments and assessment of our manuscript. Errors in line 372 corrected as well as numbers of figures referred. In addition, the error checking of the entire publication text was checked. Corrections are highlighted.
Best Regards,
Tomas Čepaitis
This manuscript is a resubmission of an earlier submission. The following is a list of the peer review reports and author responses from that submission.